# Reinforcement Based Learning on Classification Task Could Yield Better Generalization and Adversarial Accuracy

**Shashi Kant Gupta**
Department of Electrical Engineering
IIT Kanpur, UP, India
shashikg@iitk.ac.in

## Abstract

Deep Learning has become interestingly popular in computer vision, mostly attaining near or above human-level performance in various vision tasks. But recent work has also demonstrated that these deep neural networks are very vulnerable to adversarial examples (adversarial examples - inputs to a model which are naturally similar to original data but fools the model in classifying it into a wrong class). Humans are very robust against such perturbations; one possible reason could be that humans do not learn to classify based on an error between "target label" and "predicted label" but possibly due to reinforcements that they receive on their predictions. In this work, we proposed a novel method to train deep learning models on an image classification task. We used a reward-based optimization function, similar to the vanilla policy gradient method used in reinforcement learning, to train our model instead of conventional cross-entropy loss. An empirical evaluation on the cifar10 dataset showed that our method learns a more robust classifier than the same model architecture trained using cross-entropy loss function (on adversarial training). At the same time, our method shows a better generalization with the difference in test accuracy and train accuracy $< 2\%$ for most of the time compared to the cross-entropy one, whose difference most of the time remains $> 2\%$.

## 1 Introduction

There's a tremendous increase in using deep learning models for various perceptual tasks in computer vision. Thousands of new works are being published every year on attaining better accuracy on different datasets [14]. But these deep neural networks are very vulnerable to adversarial examples (adversarial examples - inputs to a model which are naturally similar to original data but fools the model in classifying it into a wrong class) [10], which raises a concern about whether the model should be used for real-time application purpose or not. While the past few years have seen intense research in training robust models against adversarial attacks, most of them have focused on using various adversarial training approaches, unlabelled data, or revisiting misclassified examples [15, 5, 21]. On the other hand, humans are very robust against such perturbations. Previously, Engstrom et al. [8] and Ilyas et al. [12] shown how adversarial perturbations correspond to non-robust but predictive features and that adversarial robustness can indeed help deep neural networks in learning perceptually similar features. This gives us a hint to look at the problem of adversarial robustness by introducing a more human-like learning approach. So we tried to find a different approach to train deep neural networks based on how possibly humans learn to classify images. Humans do not learn to classify based on an absolute error between "target label" and "predicted label". Instead, they predict a label and then get feedback from another person or a source, which results in a positive or negative reward for the learner, and based on this reward; they learn to classify. Past research has also shown that if only one

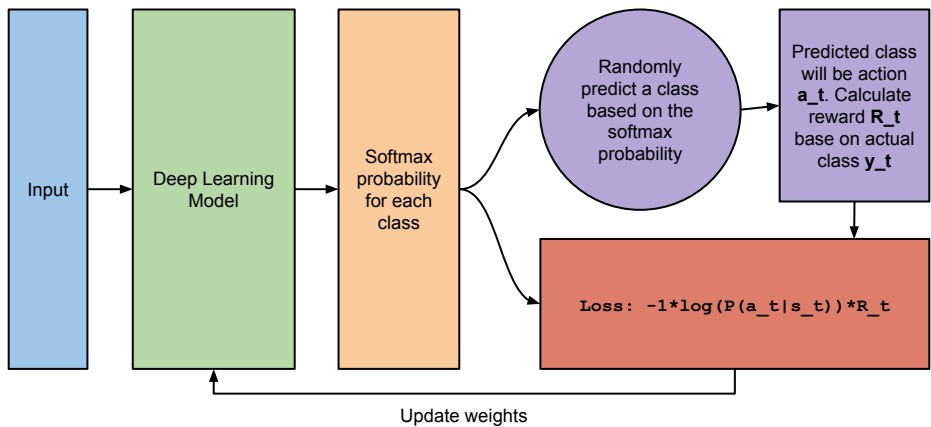

Figure 1: The complete flowchart for proposed reinforcement based classification model

type of reward is given, i.e., either a positive or negative reward, learning follows a simple rule-based approach. But if both negative and positive rewards are given, learning follows an integration-based approach [3, 2, 18].

In this work, we focused on introducing a new kind of objective function. We used a reward-based optimization function, similar to the vanilla policy gradient method in reinforcement learning, to train our model instead of conventional cross-entropy loss. Our formulation is fairly simple and similar to the vanilla policy gradient [23]. We design the reward environment, which is as simple as giving positive rewards for correct classification and negative rewards for the wrong classification. And in the end, we train the model to maximize reward using a policy gradient. Here our policy is simply the softmax probability distribution to classify the given input image among the various classes. We trained a very minimal CNN architecture against FGSM attacks [10] using our method and cross-entropy loss. We observed that even though the model trained using cross-entropy achieved higher accuracy on the test set than our approach, the accuracy against adversarial examples (generated using test data) was higher for our method. One more exciting result was that our method generalizes much better than the model trained using cross-entropy loss, i.e., our training and validation accuracy remains almost close to each other.

## 2    Method

A combination of deep learning and reinforcement learning (RL) is widely being used in state-based decision-making tasks [16, 9]. But there are very few works that have focused on reinforcement-based learning in classification tasks [22]. As per our knowledge, no evaluation is done on its adversarial robustness, and generalization. Also, previous implementations of RL for classification are more complex than ours. We propose a fairly simple implementation. We designed a simple RL environment for the classification task, which will act similar to other RL environments used to evaluate reinforcement learning algorithms. This environment can be used to run various RL algorithms to train on any classification task. In this work, we used the basic formulation of the Vanilla Policy Gradient (VPG) method to test our method [23]. In our formulation, the state $s_t$ is the input image, action $a_t$ is the predicted class, and reward $R_t$ depends upon $a_t$ and actual class label $y_t$. If $a_t = y_t$, we give a positive reward (+1) on the other hand if $a_t \neq y_t$, we give a negative reward (-1). Finally, we train the model using VPG loss (refer to Eq. 1). Here $t$ is the example image in the given training batch of size $B$.

$$\sum_t^B -\frac{1}{B} log(P(a_t|s_t)) * R_t \tag{1}$$

The complete flowchart is shown in Figure 1. Note that during training, the model takes action by using random selection based on the softmax probability, which is extensively used in RL to bring randomness during training. One direct benefit of this implementation is that we are

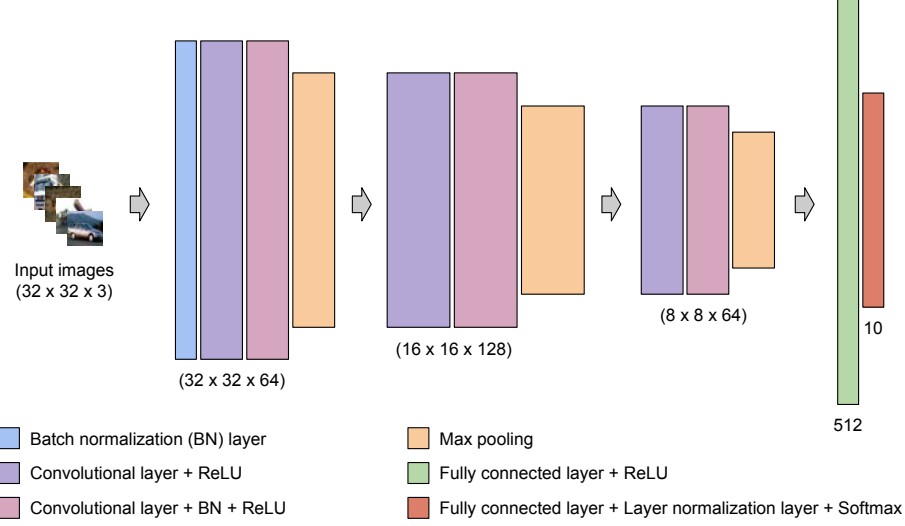

Input images
(32 x 32 x 3)

(32 x 32 x 64)

(16 x 16 x 128)

(8 x 8 x 64)

10

512

- ■ Batch normalization (BN) layer
- ■ Convolutional layer + ReLU
- ■ Convolutional layer + BN + ReLU
- ■ Max pooling
- ■ Fully connected layer + ReLU
- ■ Fully connected layer + Layer normalization layer + Softmax

Figure 2: CNN model architecture used for the study. All the convolutional layer have convolution window of size 3 x 3. All the max pooling layers have pooling window of size 2 x 2.

penalizing/rewarding our network based on what predictions it makes. So if the network predicts a wrong class, it will be penalized based on the gradients calculated using that specific wrong class, which in comparison to cross-entropy depends only on the gradients calculated using the correct label. Another advantage of using this method is that we can develop different strategies for assigning rewards based on the networks' prediction. For example, giving higher negative rewards to those examples on which the network makes repeated wrong predictions could help improve the learning.

## 3 Experiment

The CNN architecture used for the study is shown in Figure 2. The architecture follows a similar heuristic of convolution layers used in VGG16 architecture [19], but the network's depth is relatively less. One more important thing to note in the architecture we used is the "layer normalization" layer just before the softmax activation, which we experimentally found to avoid obfuscated gradient issue effectively [4]. We first performed the evaluation without any specific adversarial training approach. We found that the RL method indeed showed better FGSM adversarial accuracy than the CE method (refer to Appendices, Figure 5). But it was easy to attack the trained model via AutoAttack (an ensemble of diverse parameter-free attacks [7]). We then performed adversarial training by generating adversarial images using FGSM attack (at $\epsilon = 8/255$). The adversarial images were generated at the beginning of each training step (refer to Equation 1 in Appendices for adversarial training algorithm). We trained the model on those adversarial images using both the methods, i.e., cross-entropy (CE) one and our proposed method (RL) one for 220 epochs with RMSprop optimizer at $learning\_rate = 0.0001$ and $decay = 1e-6$ (The choice of hyperparameter was taken from TensorFlow example model on cifar10). We initially made checkpoints of our model at intervals of 20 epochs and reporting training, testing, and adversarial accuracy (on test data) at those intervals (4). We picked the checkpoint at which the respective method performed best on adversarial images generated using test data. For the RL model, it was the 220th epoch. And for the CE model, it was the 60th epoch. Finally, the robustness of both of the models was tested using AutoAttack. To test the CE method's performance at near the 60th epoch, after which the CE model's performance starts to decrease and for the sake of generalization, we retrained the model from scratch in which we recorded the training history till 150 epochs. We found similar results (Appendices, Figure 6).

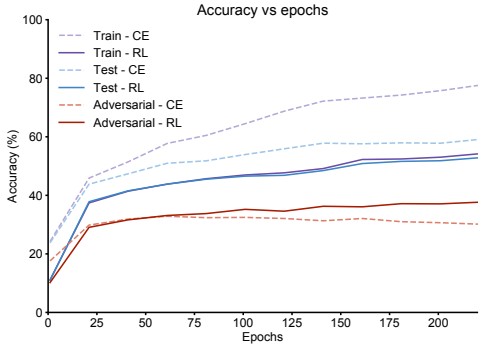
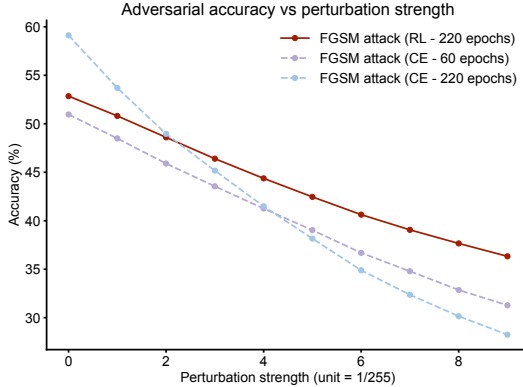

Figure 3: Performance of the trained model with time. Adversarial accuracy corresponds to FGSM attacks with $\epsilon = 8/255$ on test cifar10 test data

Figure 4: Accuracy against FGSM attacks at various perturbation strength on cifar10 test data

# 4 Results

The training performance for both the model is shown in Figure 3. The model trained using cross-entropy loss is abbreviated as 'CE' while the model trained using our reinforcement-based method is abbreviated as 'RL.' The CE model reaches its maximum adversarial accuracy at 60th epoch with adversarial accuracy on FGSM attack 32.86% and reduces afterward. In contrast, the RL model performance improved over successive epochs with adversarial accuracy on FGSM attack 37.66% at the end of 220th epoch. We can also see that the RL model generalizes much better than the CE model, with its training and testing curve to be very close to each other (Figure 3). We also tested the robustness accuracy for both the models, i.e., CE and RL one, against different perturbations level added to the original image. Results are shown in Figure 4. RL model always performed better than the CE model, which was trained till 220 epochs. We also tested the CE model, which was trained till 60th epochs; we found that for $eps \leq 3/255$, the CE model performed better than the RL model, but its performance got worse on reaching $eps = 8/255$.

We further tested the robustness of both the model using AutoAttack (contains ensemble of various adversarial attacks - consisting of apgd-ce, apgd-t, fab-t, and square attack [7, 1, 6]) to confirm the robustness against several other adversarial attacks. We found that our model still performs much better than the CE model (Table 1). Please note that all the reported accuracy are on cifar10's test data.

| Model | Natural Acc. | Adversarial Acc. |
|---|---|---|
| RL (220th epoch) | 52.85% | 30.41% |
| CE (60th epoch) | 50.96% | 26.36% |
| CE (220th epoch) | 59.12% | 17.95% |

Table 1: Accuracy of both the models against AutoAttack (eps = $8/255$) on test data

There are some concerns regarding the issue of label leaking with the FGSM method for adversarial training [13]. Therefore we also evaluated the performance CE case with adversarial training using PGD-Linf attack (eps = 8/255, step_size=2/255, and num_step=5)[15]. We show that even if we train the network using cross-entropy loss on adversarial images generated using PGD-Linf attack, the RL method's performance remains better than the CE method. The best accuracy against AutoAttack for CE method trained using PGD-Linf attack on test data was 27.76%. Moreover, some recent work suggests that the FGSM based method can be used for adversarial training with proper early stopping [24].

## 5  Discussion

In this work, we proposed a fairly simple method to introduce reinforcement-based learning for classification tasks. Our method shows an initial sign of improvements for better generalization and more robustness to adversarial attacks than the same model architecture trained on cross-entropy loss. Even though our present model does not beat the SOTA results if we consider some recent work involving faster training for a robust model against adversarial attack. Our model performance is quite comparable and outperforms [20] which shows 29.35% accuracy on cifar10 against AutoAttack [7]. But considering that this is a preliminary result, we still need to evaluate our method on a larger scale with a more complex model known to perform best on cifar10 and better parameter tuning.

On the other hand, our RL formulation for classification task could probably benefit other computer vision problems such as datasets having imbalanced class in which we can change the magnitude of positive or negative reward for different classes to remove discrepancy due to imbalanced data. For future research, it would also be interesting to evaluate how the model performs if we change the reward assignment method. It would also be interesting to test model trained using RL method on predicting human similarity judgments (Peterson et al. [17]).

Finally, we hope that this result will bring on exciting discussions on the intersection of cognitive sciences and artificial intelligence in understanding how humans might be learning to classify different objects.

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

# Appendices

## A. Adversarial Training Algorithm

The pseudo-code shown below does not show the FGSM attack algorithm [10] and RMSprop [11]. Both are very widely known, and their algorithms can be referred from the original work.

---

**Algorithm 1:** FGSM adversarial training for $T$ epochs; perturbation strength of $\epsilon$; batch size of $B$; data size of $N$; x is the overall dataset; y is labels for the dataset; $f_\theta$ is the network; $\theta$ is weight of the network

---

**for** $t = 1, 2, \ldots T$ **do**

    $x_{shuffle} = shuffle\_dataset(x)$ ;

    **for** $b = 1, 2, \ldots N/B$ **do**

        $x_b, y_b = x_{shuffle}[(b-1)*B : b*B], y[(b-1)*B : b*B]$ ;

        $x_{adv} = FGSM\_attack(x_b, y_b, \epsilon)$ ; // `perform FGSM attack`

        `// Calculate` $\nabla\theta$ `using RMSprop optimiser`

        $\nabla\theta = RMSprop\_optimiser(loss(f_\theta(x_{adv}), y_b), \theta)$ ;

        $\theta = \theta - \nabla\theta$ ; // `update` $\theta$

    **end**

**end**

---

## B. Results Without Any Adversarial Training Against FGSM Attacks

We also performed the evaluation without any adversarial training against FGSM attack for which the RL model indeed showed better FGSM adversarial accuracy than the CE model. But as stated before, AutoAttack very easily attacked this.

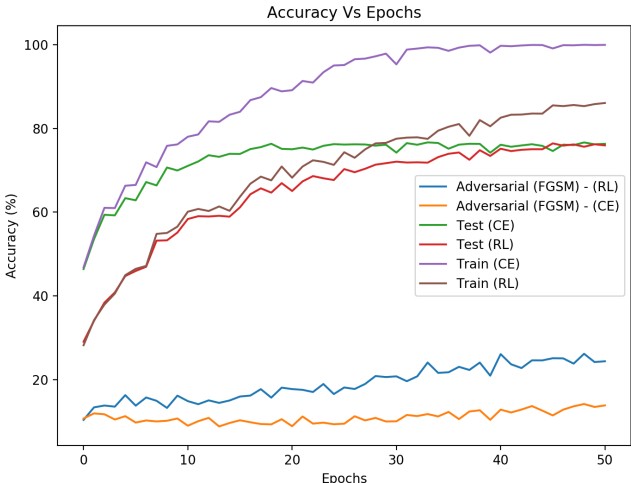

Figure 5: Results without adversarial training. We can see RL method still performs better than the CE method. Adversarial accuracy (FGSM) is calculated for $eps = 8/255$

**C. Adversarial Accuracy vs Epochs on Another Training Instance (with adversarial training)**

We initially made checkpoints of our model at intervals of 20 epochs. To test the model's performance at near the 60th epoch, after which the CE model's performance starts to decrease and for the sake of generalization, we retrained the model from scratch in which we recorded the training history till 150 epochs. It again resulted in similar results. Results are shown in Figure 6.

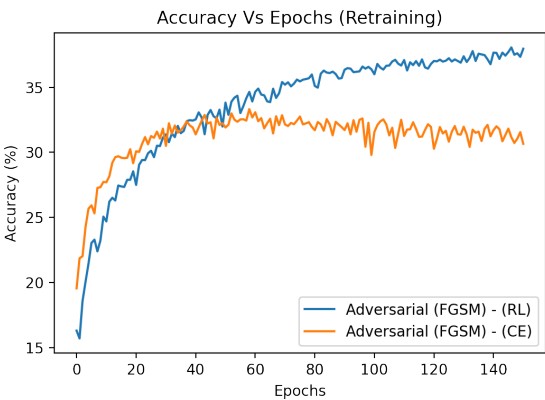

Figure 6: Adversarial accuracy on cifar10 test data (FGSM) vs epochs for $eps = 8/255$

# D. Examples of Gradient Calculated on Models Trained Using Both Methods

To look further into the kind of features learned by both types of the objective function. We checked the gradients calculated during the FGSM attack on both the models. While for many of the examples, we did not find any useful differences between the two methods, some of the examples were worth adding here in which we did find the difference between the two methods. A closer look into various examples is needed to make any clear distinction.

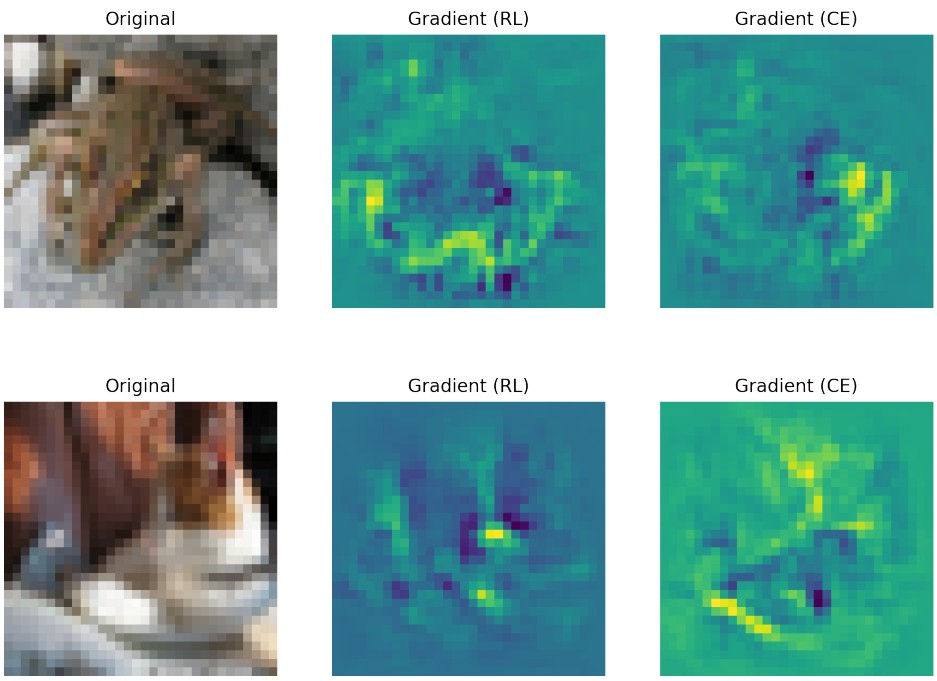

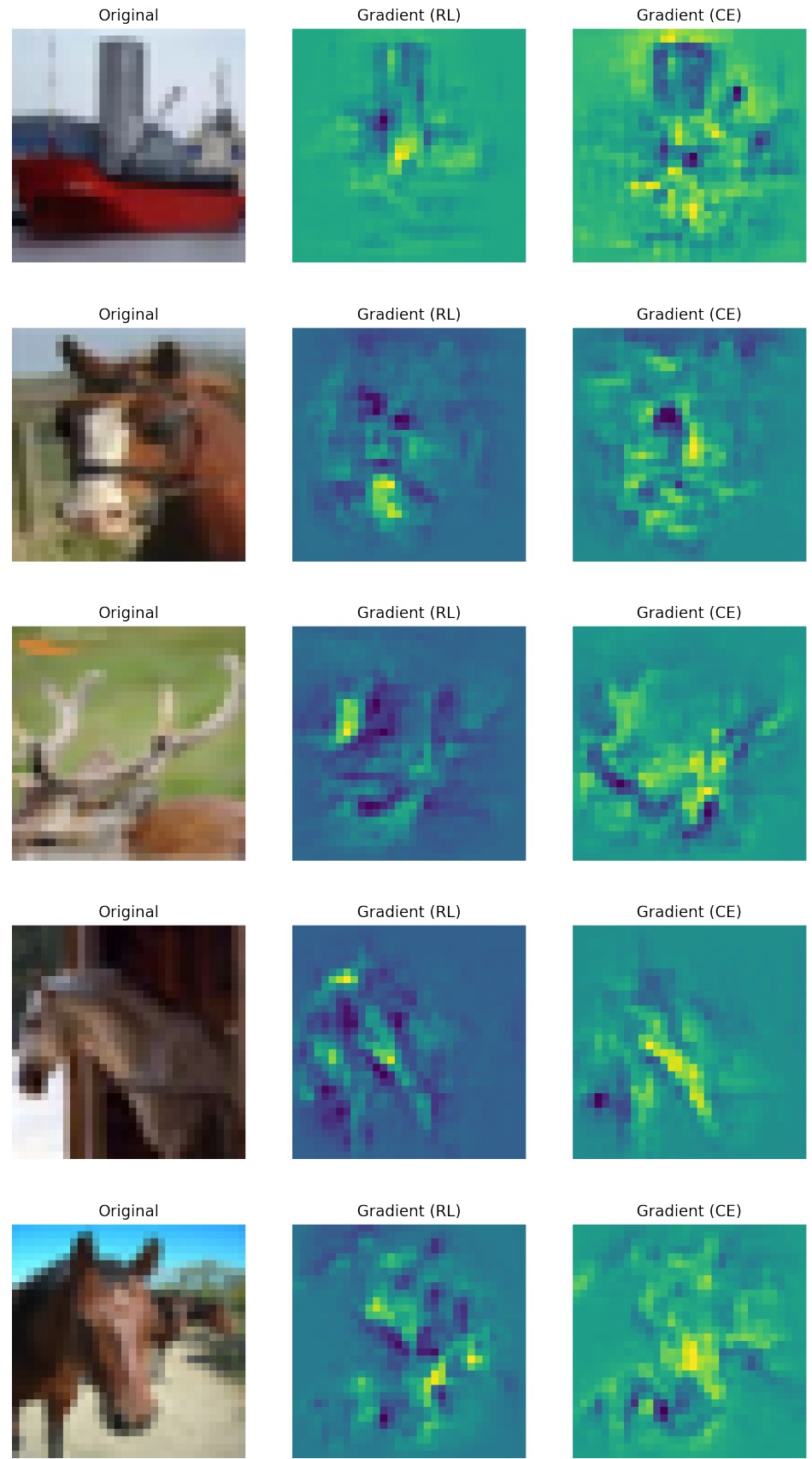

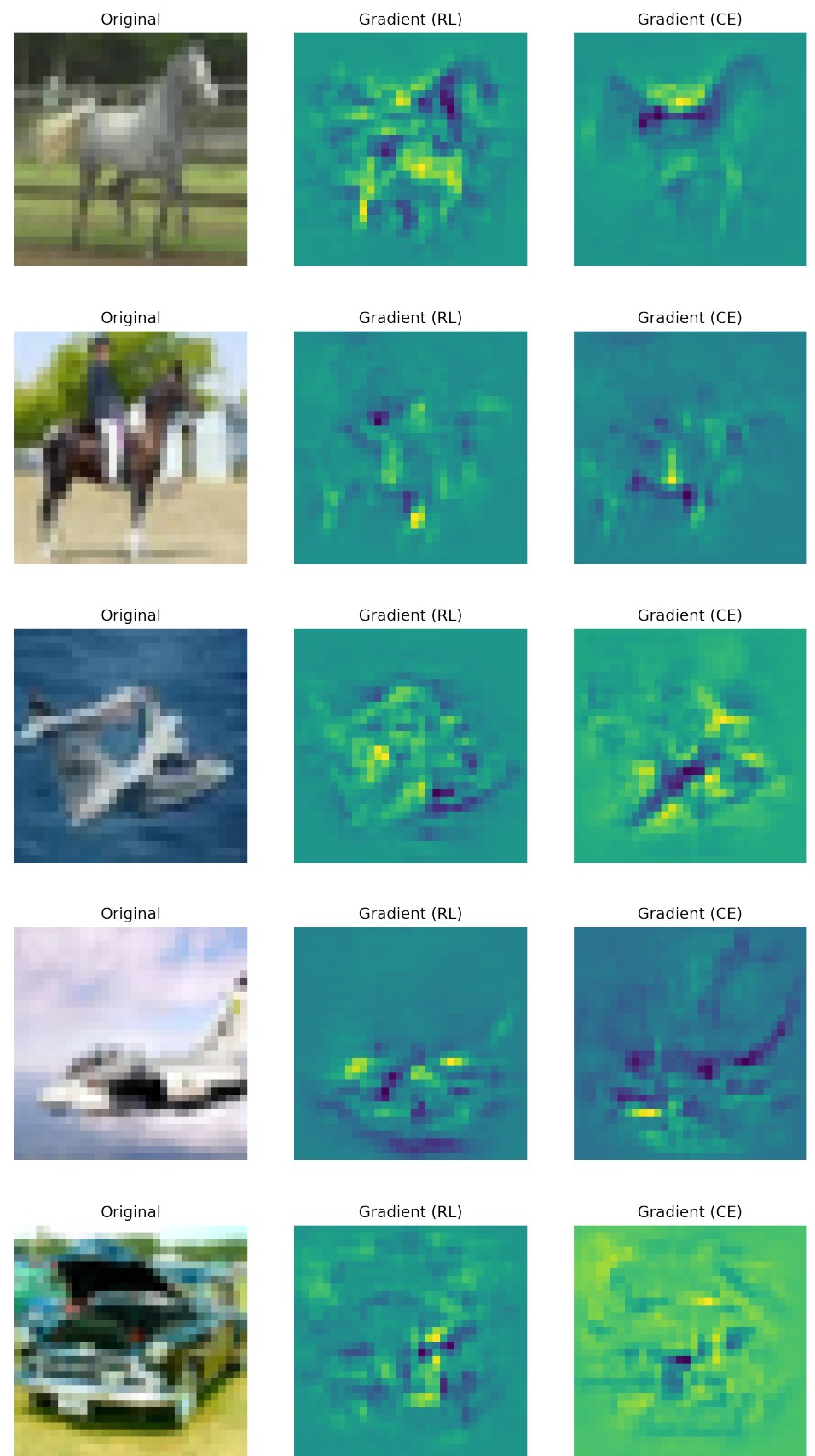

