# OpenReview forum: "Reinforcement Based Learning on Classification Task Could Yield Better Generalization and Adversarial Accuracy"
_NeurIPS.cc/2020/Workshop/SVRHM — SVRHM@NeurIPS Poster_

### Official Review · AnonReviewer2 · 2020-10-28
**Review for Reinforcement Based Learning on Classification Task Could Yield Better Generalisation and Adversarial Accuracy**

**Rating:** 7
**Confidence:** 3

**Review:**

This work investigates using a policy loss from Reinforcement Learning (RL) to train an image-classifying CNN on CIFAR10. The authors suggest that an RL policy loss may be closer to the way humans learn compared to crossentropy. This network shows some improvements in robustness compared to a network trained using the classic crossentropy loss -  it has higher adversarial robustness and less difference between training and testing error. However, it does not quite reach performance level of the crossentropy trained network on non-adversarial images.

I think this approach is interesting and fits the SVRHM workshop well. The experiments are generally well done. Therefore, I suggest accepting this contribution. I have some suggestions to potentially improve the paper.

-	None of the networks, including the crossentropy one, surpass 60% test accuracy on CIFAR10. Why is this? The state of the art is 1-2% error rate, and to my understanding it is easy nowadays to reach at least 80%. This low performance may suggest that the crossentropy network trains suboptimally, which would question the results in general. Can the authors explain this low performance and show that it is not a problem? If it comes from the network architecture, why not choose a better performing architecture?
-	There is “individual variability” in deep learning, i.e., the initial weights of a network impact performance. This is especially true in deep RL. There is a concern that the initialization, and not the training procedure, might drive the results here. Comparing networks with the same random seed would fix this problem. Alternatively, several networks could be trained and the average & standard deviation of the performance could be plotted.
-	The authors suggest that their approach might be closer to how humans learn. Is there any scientific evidence for this?
-	The clarity of presentation could also be improved.

---

### Official Review · AnonReviewer1 · 2020-10-30
**Interesting idea but experiments may need to be improved**

**Rating:** 5
**Confidence:** 4

**Review:**

This paper examines the generalization properties and adversarial robustness in CNNs trained via a reinforcement learning method (vanilla policy gradient) with those trained using the cross-entropy loss. The paper reports higher adversarial accuracy and smaller generalization gap in the neural network model trained with reinforcement learning.

Complementing supervised training of neural networks with RL-based methods is a promising approach. However, I have some concerns with the way the experiments are conducted in this paper and I would like to see them resolve before this paper is presented in this venue.

My main concern is regarding the control CE model trained on FGSM attack. The FGSM attack has been shown to be unsuitable for adversarial training (AT) due to "label leaking" [1]. Current AT methods unanimously advise to train on iterative attacks like PGD-Linf. My concern is also amplified by the results reported in Table-1 where the CE model's adversarial performance seems to be getting substantially worse with more training.

Other comments/concerns:

It is claimed that: "Humans do not learn to classify on the basis error between "target label" and "predicted label" but instead they predicts a label and then some other people give them feedback which results into a positive or negative reward for the learner and on the basis of this reinforcements they learn to classify different stimuli." The true objective and learning algorithm in the human brain is being debated and NOT known. RL is only one of these mechanisms and potentially not the primary one in the sensory cortices.

line79: How was the checkpoint picked? My impression is that the checkpoints might have been selected based on the test accuracy which will be incorrect. In any case, this should be clarified here.

Several statements in the abstract are unclear:
- "An empirical evaluation on cifar10 dataset showed that our method outperforms the same model architecture trained using cross-entropy loss function" --> outperformed at what?
- "At the same time, our method generalizes better to the training data" --> do you mean generalizes better to the TEST data?

The paper needs better proof-reading, there are more than few typos or grammatical mistakes in the text. e.g. in abstract: "on the basis error" --> "on the basis of error", line68: fasten --> improve?

[1] Kurakin, Alexey, Ian Goodfellow, and Samy Bengio. "Adversarial machine learning at scale." arXiv preprint arXiv:1611.01236 (2016).

---

### Official Review · AnonReviewer3 · 2020-11-01
**Review of "Reinforcement Based Learning on Classification Task Could Yield Better Generalisation and Adversarial Accuracy"**

**Rating:** 6
**Confidence:** 4

**Review:**

Here the authors show that using a reward-based optimization function in a categorization task can lead to better performance of a DNN with adversarial examples than more traditional approaches. The paper is relatively clear, and the result is rather modest but interesting. I only have a couple of comments about how this paper/poster could be improved:

- The writing is very unclear, full of typos and grammatical errors. The paper needs to be thoroughly proofread before acceptance. Examples in lines 32, 33, 35, 51, 68, 98, 167, 169, 171, 172, 173.

- Figure 2 is not useful to understand the model architecture. It consist of mostly text and full of terms (acronyms? function names?) that are not explained in the figure caption. A better graphical representation of the model architecture, together with explanation of terms in the caption would improve the paper greatly.

- The final result is rather modest, and there is no discussion of the prior relevant literature combining RL with deep learning. For example, some researchers have used deep learning for state representation learning in RL (e.g., Bohmer et al., 2015; de Bruin et al., 2018). Because the authors do not offer links to that literature, it is not clear how their approach relates to those previous efforts and to what extent it is novel.

---

### Public Comment · ~Shashi_Kant_Gupta1 · 2020-12-08
**Response to reviewers and update on camera ready version**

We thank our reviewers for providing in-depth insights into our paper. Based on their feedback and suggestions, we have made some changes to the camera-ready version of the paper. And we will certainly implement other suggestions in our future work. Below is a brief detail of major changes and our reply to their concerns:

**Changes to papers**
1. We corrected the typos and grammatical errors (R1 and R3)
2. Replaced the figure for model architecture and provided detailed figure captions (R3).
3. We have added references and discussion on prior works in Deep RL (R3) and human learning & reinforcement (R2)
4. A brief discussion on label leaking with FGSM and a cross-check on PGD-Linf attack (cross-entropy case) (R1)
5. Clarification on how checkpoint is selected (R1)

**Comments on major concerns**

**Label leaking:** First of all, we completely understand the reviewer’s concern, and therefore, we evaluated the performance of cross-entropy case with adversarial training using PGD-Linf attack (eps = 8/255, step_size=2.0, and num_step=5). We picked the checkpoint based on the best adversarial accuracy on test data. The accuracy of the trained model against AutoAttack [1] on test data was 27.76%, which is still low than the accuracy achieved by the RL case (30.41%). We would also like to clarify that label leaking is said to occur when the accuracy on the clean dataset is poorer than its perturbed dataset [2], which is not the case with our results. FGSM method can be used for adversarial training; a more comprehensive evaluation is done in this paper [3]. Moreover, PGD based attacks are so time-hungry, if RL based methods can solve that issue by using one-step FGSM attacks, then the main objectives of the paper remain intact.

**Checkpoints:**  We have added clarification regarding this in the updated version of the paper. We disagree with the reviewer that using the test data will be incorrect here. The experiment’s motivation is not to train a model for deployment purposes but rather to test which method works best. Therefore, we think it's okay to chose checkpoints based on test data.

**Cross entropy network trains suboptimally:** Due to resource limitations, we could not test the method on more complex architectures for now, but we will certainly include these in our future works. We would also like to point out that generally, adversarial training decreases standard accuracy [4]. In our supplementary, we have also shown the network’s performance without any adversarial training (the test set accuracy for the network reaches ~ 75 % in 50 epochs).

**Individual variability:** We verified this by training the model using a random seed; no difference was observed in the paper’s conclusion. We considered the point for reporting mean and standard deviation to be very important and will definitely include that in our future work.

**References**

[1] Croce, F., & Hein, M. (2020). Reliable evaluation of adversarial robustness with an ensemble of diverse parameter-free attacks. arXiv preprint arXiv:2003.01690.

[2] Kurakin, A., Goodfellow, I., & Bengio, S. (2016). Adversarial machine learning at scale. arXiv preprint arXiv:1611.01236.

[3] Wong, E., Rice, L., & Kolter, J. Z. (2020). Fast is better than free: Revisiting adversarial training. arXiv preprint arXiv:2001.03994.

[4] Tsipras, D., Santurkar, S., Engstrom, L., Turner, A., & Madry, A. (2018). Robustness may be at odds with accuracy. arXiv preprint arXiv:1805.12152.

---

### Decision · Program_Chairs · 2020-11-02

Accept (Poster)